# The Progress of Advanced Ultrasonography in Assessing Aortic Stiffness and the Application Discrepancy between Humans and Rodents

**DOI:** 10.3390/diagnostics11030454

**Published:** 2021-03-06

**Authors:** Wenqian Wu, Mingxing Xie, Hongyu Qiu

**Affiliations:** 1Center for Molecular and Translational Medicine, Institute of Biomedical Science, Georgia State University, Atlanta, GA 30303, USA; wwenqian@gsu.edu; 2Department of Ultrasound, Union Hospital, Tongji Medical College, Huazhong University of Science and Technology, Wuhan 430022, China

**Keywords:** aortic stiffness, atherosclerosis, ultrasonography, hemodynamics, cardiovascular risk factor

## Abstract

Aortic stiffening is a fundamental pathological alteration of atherosclerosis and other various aging-associated vascular diseases, and it is also an independent risk factor of cardiovascular morbidity and mortality. Ultrasonography is a critical non-invasive method widely used in assessing aortic structure, function, and hemodynamics in humans, playing a crucial role in predicting the pathogenesis and adverse outcomes of vascular diseases. However, its applications in rodent models remain relatively limited, hindering the progress of the research. Here, we summarized the progress of the advanced ultrasonographic techniques applied in evaluating aortic stiffness. With multiple illustrative images, we mainly characterized various ultrasound techniques in assessing aortic stiffness based on the alterations of aortic structure, hemodynamics, and tissue motion. We also discussed the discrepancy of their applications in humans and rodents and explored the potential optimized strategies in the experimental research with animal models. This updated information would help to better understand the nature of ultrasound techniques and provide a valuable prospect for their applications in assessing aortic stiffness in basic science research, particularly with small animals.

## 1. Introduction

Cardiovascular diseases remain one of the leading causes of morbidity and mortality, drawing numerous clinical and basic science studies. Various techniques have been clinically developed to detect the structural and functional alterations of cardiovascular systems, including non-invasive methods such as ultrasonography, magnetic resonance imaging, computed tomography, positron emission tomography, and single-photon emission computed tomography, and invasive procedures such as cardiac catheterization [1,2,3,4]. The applications of these new inventions have dramatically changed our knowledge and understanding of cardiovascular diseases and improved the evaluation of the latest treatments’ effects. Importantly, these techniques also provide new strategies for basic science research in various animal models. While many new techniques are developing, ultrasonography remains an essential and principle tool in assessing structural and functional alterations of the cardiovascular system in humans [5,6]. However, its applications appear relatively less in vascular diseases, particularly in small animal models, due to the technique challenge.

The aorta is the largest artery that carries blood from the left ventricle (LV) to the systemic circulation [7]. Aortic stiffening is a fundamental pathological alteration of atherosclerosis and other various aging-associated vascular diseases. It is also known as an independent risk factor of cardiovascular complications and has been considered as a predictor of cardiovascular outcomes in clinical [8,9,10]. As a non-invasive method, ultrasonography has been used in clinical diagnosis for aortic stiffness in many aortic disorders such as atherosclerotic degeneration and aortic aneurysms through measuring the wall-thickness, diameter, structural defects, blood flow velocity, and other pathological changes in the aorta [7,11,12,13]. The developed techniques used in patients also provide powerful tools for basic science research in different animal models, particularly the large animal models such as monkeys, swine, canines, and horses [14,15,16,17,18,19]. The combination of ultrasonography with other invasive strategies in large animal models provides valuable direct evidence of the pathogenesis of the diseases that may not be able to obtain from human studies, which, in turn, led to the translational development of new diagnostic techniques and therapeutic strategies for predicting and preventing cardiovascular disease complications in humans [20].

Small animals, especially rodents (rats and mice), have also been widely used in the basic science research of vascular diseases, particularly in mechanistic molecular studies with the advantages of various genetic-modified models [21]. These small animal models have the feasibility to perform analysis that is not practical in humans, such as surgery, genetic and pharmacological inventions, invasive measurements, and tissue/cell isolations. Although ultrasonography has been widely used in rodents for detecting heart alterations [20,22,23,24], its application in rodent blood vessels has encountered tremendous challenges due to the specific location and remarkable smaller size of the blood vessels, as well as the rapid changes caused by the high heart rate (around four to five folds the human heart rate). Since the evaluation of vascular structure and function is crucial in determining the mechanistic regulation and pharmacological efficiency in rodents, it is essential to establish an optimal ultrasonographic assessment of the aortic alterations in these small animal models.

In this review, we summarized the progress of the ultrasonographic applications in assessing aortic stiffness and outlined the application discrepancy of the current techniques in humans and rodents. Specifically, using multiple illustrative images from both human and mouse models, we characterized the nature of each ultrasound technique in the assessments of aortic stiffness based on the alteration of aortic structure, hemodynamics, and tissue motion. We also discussed the potential optimized strategies to improve the applications of these techniques in rodents. The updated information would provide a valuable prospect for the application of clinical approaches in basic science research.

## 2. Integrating Applications of Ultrasonography in the Assessments of Aortic Stiffness

The aorta consists of five anatomical segments from the proximal to distal ends, including the aortic root (AoR), ascending aorta (AAo), aortic arch (AoA), descending thoracic aorta (DTAo), and descending abdominal aorta (DAAo). The ultrasonographic images of the aorta can be obtained with multiple views and have been used to assess the aortic structural and functional alterations in humans and large animals as well as in rodents.

Aortic stiffness is a combined phenotype that could be changed by the long-term remodeling of the wall constituents or with the acute dysfunction of smooth muscle and/or endothelial cells. Ultrasonographic evaluations of the aorta could be interrogated from various imaging [7], including (a) real-time B-mode images and M-mode traces; (b) doppler ultrasonography [25]; (c) pulse wave velocity (PWV); and (d) newly developed processing technologies, such as speckle tracking imaging (STI), power mode imaging, elastography, and contrast-enhanced ultrasound [26,27,28]. Due to the complexity of the aortic mechanical properties and functions, no signal parameter could represent the full characterization of aortic stiffness; thus, the assessment of aortic stiffness usually requires a combination of multiple measurements and analysis by integrating various ultrasonography. As summarized in Figure 1, aortic stiffness could be assessed through the following three essential measurements based on aortic structure, hemodynamics, and tissue motion, respectively.

### 2.1. Structure-Based Assessment of Aortic Stiffness

Stiffness refers to the resistance to deformation. Thus, aortic stiffness can be assessed based on its capability for structural alteration, such as luminal diameter and vessel wall thickness, which can be accurately measured by M- and B-mode ultrasonography.

#### 2.1.1. Image Acquisition of Aorta by M- and B-Mode Ultrasonography

M-mode ultrasonography detects the structures along a single axis, by which the tissue interfaces are represented in dots on the display screen. With a high temporal resolution, M-mode images can measure the inner diameter and wall thickness of the aorta from several cardiac cycles. Thus, systolic and diastolic diameters could be obtained simultaneously [29]. As illustrated in Figure 2, an M-mode line can be placed perpendicular to the aortic walls, such as a long-axis view of the AAo (Figure 2A,B) and a short-axis view of the DAAo (Figure 2C,D). The image acquisition of the aorta by M-mode ultrasonography and its application in assessing aortic dimensions are similar between humans (Figure 2A,C) and rodents (Figure 2B,D).

B-mode ultrasonography, also known as two-dimensional (2D) ultrasonography, is one of the most basic ultrasound models that produces a real-time black/white image of the targeted site, where the aortic wall is shown as echo-reflecting and the lumen as echo-free. As shown in Figure 3, the ultrasonographic view of the aorta is either circular in the perpendicular sections or tube-shaped in the parallel sections in humans. For example, the AoR and the AAo could be visualized in the parasternal long-axis view (Figure 3A), the modified right parasternal long-axis view, modified apical five-chamber (Figure 3B), and three-chamber views (Figure 3C). Additionally, the suprasternal view is a crucial view to visualize the AoA and the three supra-aortic trunks (innominate, left carotid, and left subclavian arteries), and a variable tract of the AAo and the DTAo (Figure 3D). In some cases, the entire arch could not be visualized in a single image plane because of the aorta’s extreme tortuosity. The DTAo can also be displayed in the posterior field through the cardiac acoustic window (Figure 3E). Moreover, parts of the DTAo may be invisible due to the tracheal air. The subcostal views may be helpful and allow the DAAo to be visualized (Figure 3F). Although B-mode has been widely used in humans, its application in mouse aorta remains relatively challenging, primarily because of the difficulty in obtaining exceptional spatial and temporal resolution in a small, rapidly varying vessel. Since apical views of the heart are prone to change in rodents, particularly in mice, it is difficult to obtain stable images of the aorta in this view. Reciprocally, the parasternal (Figure 4A), suprasternal (Figure 4B), subcostal, and transabdominal views (Figure 4C) are the most critical views for murine aorta, which enable an adequate assessment of the AoR, the AAo, and most of the segments of the DTAo and DAAo in mouse models.

#### 2.1.2. Parameters of Aortic Stiffness Assessment by M- and B-Mode Images

Although current clinical guidelines support the use of aortic diameter to predict the complications and guide elective aortic surgery [30], the aortic diameter per se may be not suitable to determine the early functional abnormality of the aortic wall due to the segmental differences along with the aorta [31,32] and the variations in pathophysiologic changes of aortic tissue [26]. Therefore, a series of combined parameters were established to assess the aortic mechanical property in patients and animal models based on the alteration of local aortic diameters during the systolic and diastolic cycles or under different blood pressure (BP).

As shown in Table 1, the structural measurements of the aorta with M- and B-mode images have been used to determine comprehensive parameters of aortic stiffness or vascular elasticity. For example, aortic strain (AS), calculated by a few formulas based on the alterations of aortic systolic and diastolic diameters, was used to reflect a deformation of the vessels under cardiac cycles, such as AS = (AoS − AoD)/AoD or circumferential strain = ½ × [(AoS/AoD)^2^ − 1] × 100%, while AoS and AoD represent the systole and aortic diameters, respectively. In addition, as aortic stiffness is pressure-dependent, stiffness parameters were calculated by various formulas based on the changes in the area (or in diameters) for a given pressure step (ΔP). For examples, arterial compliance (C) was calculated by the ratio of the absolute change in diameter (ΔD = AoS − AoD) and pressure (ΔP = SBP − DBP) at a fixed vessel length (C = ΔD/ΔP), and Young’s modulus (E_Y_) was calculated by AoD/H/(ΔA/A × ΔP), while H represents aortic wall thickness, A represents the minimal cross-sectional area of the aorta and ΔA as the maximal minus minimal cross-sectional area of the aorta. By contrast, some other indexes were calculated by the stress/strain ratio, such as the elastic pressure-strain modulus (Ep = 1333 ΔP/AS); aortic stiffness index (SI) = ln ΔP/AS; β index = ln (SBP/DBP)/AS), or the distensibility (DI) by (2 AS/ΔP), which is the inverse of Ep. Furthermore, strategies were also used to determine the vessel’s response to the wide changes of BP with drug administration. The correlation between the diameter changes or the wall thickness to the BP was used to reflect the aortic stiffness [14,15]. The relevant formulas of the aortic stiffness parameters using M- and B-mode and the citations are summarized in Table 1.

#### 2.1.3. Applications of M- and B-Mode Images in Aortic Stiffness Assessments

M-mode images have been used in detecting aortic stiffness in humans in both normal tissue development and pathobiological vascular diseases. For example, M-mode was used to determine the aortic elasticity during the development of isolated bicuspid aortic valve (BAV) and healthy children. By calculating the strain, DI, and SI of AAo, the results showed that children with BAV exhibited abnormal aortic elasticity from infancy to adolescence [33]. In addition, Rosca et al. showed for the first time that, in patients with severe aortic stenosis, the increased aortic rigidity, as assessed by aortic β index, was independently correlated with reduced LV longitudinal function and increased LV filling pressures as well as B-type natriuretic peptide levels. These results revealed that the increased aortic stiffness could potentially be used for prognostic prediction in patients with aortic stenosis [42].

In addition, many studies have used the parameters calculated from B-mode images to evaluate the mechanical property of the aorta, including the aortic size index (ASI), a ratio of aortic diameter and body surface area, or aortic root z-score [9,45,46]. It has been shown that ASI is positively correlated with the incidence of aortic rupture and dissection or their associated death [45]. The results showed significantly larger aortic dimensions and z-score values in the children with BAV than the control group, and a z-score >2 indicated an aortic dilatation [33]. Furthermore, with B-mode images, a study in a large population of adolescents and adults also showed an independent association of AoR size with age, body size, and gender; i.e., AoR diameter is more extensive in men than women and increases with body size and age [47]. Importantly, studies have shown that B-mode images presented a higher reproducibility in the measurements of Valsalva’s sinuses in patients with suspected Marfan syndrome [48]. B-mode has been applied to the analysis of arterial distension of the carotid arteries and the DAAo in preclinical studies of disease [21]. Measurements of arterial stiffness at different vascular tree sites do not seem to be interchangeable, but there is a correlation [49].

Moreover, a structure-based aortic stiffness assessment has been applied in large animal models to determine the regional aortic stiffness. A recent study used B-mode-derived AS to measure the aortic stiffness at two different transverse sections (renal and iliac level) with or without hypertension and indicated that the regional AS can be used to assess abdominal aortic stiffness, especially when the indirect BP measurements are inaccurate [19]. In addition, using an invasive pressure catheter or an implanted aortic pressure probe, the regional aortic pressure can be measured directly or continuously in large animal models, which provided a more accurate and reliable assessment of regional aortic stiffness [14,15]. These methods were also used and confirmed in our previous studies in hypertensive rats in which a significantly increased local aortic wall stiffness was detected in spontaneously hypertensive rats compared to Wistar Kyoto rats, evidenced by lower arterial compliance and arterial strain [22,23,24].

Furthermore, arterial distension has also be used as an early marker for detecting arterial disease in small animal models. It has been shown that the reduced circumferential strain is correlated with an increased elastin fragmentation in fibrillin-1 hypomorphic mice [44]. In addition, a novel semi-automatic method for tracking the changes of vessel lumen diameter with B-mode images was also applicable in mouse models of vascular disease by measuring the arterial diameter over the entire B-mode cine loops [21].

#### 2.1.4. Potential Limitations of the Structure-Based Aortic Stiffness Assessment

Although M-mode imaging is technically feasible and reproducible, it provides only a one-dimensional semiquantitative assessment. Attention needs to be paid to the potential bias caused by the shift field of view. It is also critical to obtain an image perpendicular to the long aortic axis to avoid overestimating the diameter of the short aortic axis [25].

The B-mode image has the capability of determining spatial pathological alterations directly. It provides excellent value for aortic stiffness calculations due to its geometric dependencies in evaluating the aortic lumen, diameter, and wall thickness. When using B-mode images, narrowing the window of interest is crucial to obtain the highest frame rate [6]. The impacts of imaging modality on the aortic diameter measurements need to be considered. It has been shown that the anteroposterior AoR diameter at end-diastole is one to two millimeters smaller with M-mode ultrasonography than the measurements obtained by B-mode, indicating the importance of multiple planes (sagittal, coronal, axial) in B-mode ultrasonography [12].

A combination of an invasive measurement of the pressure and an M-mode measurement of the diameter could enhance the reliability in local aortic stiffness. It is notable that when assessing non-invasive circumferential strain, this should be done with short or long axis M-mode images [25].

### 2.2. Hemodynamic-Based Assessment of Aortic Stiffness

Aortic stiffness may change the blood flow resistance, which alters hemodynamics and leads to a difference in blood flow velocity. Thus, aortic stiffness can also be determined by the local hemodynamic parameters, which can be obtained by Doppler ultrasonography, such as color Doppler (CD), spectral Doppler (SD), and PWV. While both CD and SD imaging can be applied to calculate flows at different regions of vessels, PWV was widely used to quantify systemic arterial stiffness noninvasively.

#### 2.2.1. The Nature of CD/SD Ultrasonography and PWV

Doppler ultrasonography employs the Doppler effect to generate imaging of the movement of tissues or body fluids (usually blood) and their relative velocity to the probe. By calculating the frequency shift of a particular sample volume, its speed and direction can be determined and visualized. There are several applications with Doppler ultrasonography, such as CD, SD, and tissue Doppler images (TDI). While CD ultrasonography provides an image showing blood flow by the bright areas of aliasing (Figure 5A–C) and turbulence in the aorta’s narrowed segment, SD imaging provides quantitative data on blood velocities, enabling flow volumes and pressure gradients to be calculated (Figure 5D–F). An important characteristic of the waveform is the peak systolic velocity, which increases as the luminal diameter decreases, such as in the setting of stenosis. Vessel tortuosity and branching can lead to either focally increased or decreased velocity, resulting in a flow disturbance or even flow reversal [50]. These characters of Doppler images are similar in humans (Figure 5A–F) and rodents (Figure 6A–D).

PWV is defined as the speed of a pulsatile blood wave that travels along an artery. It was considered the “gold standard” method for the non-invasive measurement of aortic stiffness in humans due to its strong correlation with the risk of cardiovascular events [41,51,52]. As illustrated in Figure 7A,B, PWV is determined by the temporal shift among the pulses, e.g., PWV = D (meters)/Δt (seconds), while the transit time (Δt) refers to the time of travel of the “foot” of the wave over a known distance (D). The transit time between two arterial sites can be measured invasively with one double-sensor [53] or two single-sensor pressure catheters [54], or noninvasively via sphygmomanometer [55], magnetic resonance imaging [1,56], or ultrasonography [39,57]. In humans, PWV is usually assessed using the “foot-to-foot” velocity method from various waveforms, and the most widely used proxy for aortic PWV is carotid-femoral PWV, with transit times assessed from signals measured at the carotid and femoral arteries (Figure 7A). Reciprocally, different from humans, PWV was measured regionally in mice due to the nature of the small size of the mouse body. As shown in Figure 7B, mouse PWV was determined by the distance and the time delay of the pulse waves detected from the aorta’s proximal and distal sites.

#### 2.2.2. Applications of CD/SD and PWV in Aortic Stiffness Assessments

Whereas strain, compliance, and distensibility are local markers of arterial elasticity, aortic stiffness can also be assessed through a systemic or regional functional measurement over a certain arterial length. When the aortic stiffness progresses to an advanced stage, the blood flow in the lumen may accelerate or even disappear, which can be detected by CD imaging and quantitated by SD ultrasonography. Doppler methods have been used to characterize and evaluate the vascular remodeling that occurs following transverse aortic constriction [58]. The magnitudes and waveforms of blood velocities from both cardiac and peripheral sites are similar in mice and humans [58].

PWV was widely used to quantify both regional and systemic arterial stiffness noninvasively. Particularly, aortic PWV has been commonly used in humans to evaluate aortic stiffness and has been proved to be an independent predictor of outcome in various populations [59]. For example, a large body of evidence demonstrated that the carotid-femoral PWV is positively associated with the incidence of human cardiovascular disease [43]; this parameter has been widely used in epidemiological studies in predicting aortic stiffness-associated cardiovascular events. Styczynski et al. also compared the efficiency of PWV measurement among different methods in assessing aortic stiffness and found a close correlation between echo-PWV measurements and invasive assessment. The results suggested that Doppler ultrasonography is a reliable method of aortic PWV measurement [51]. PWV has been used to evaluate aortic stiffness in patients with Kawasaki disease, and the result showed a faster aortic PWV in patients with Kawasaki disease, suggesting an increased arterial stiffness [41].

PWV is also used to evaluate the aorta in mice. L. Lee et al. gave the first demonstration of the direct measurement of PWV noninvasively in the AoA of Marfan syndrome mice. They found that PWV was significantly increased with age in these mice [57]. PWV was also found to be increased dramatically in Mcoln1^-/-^ mice compared to their wild-type littermates, and Vitamin D treatment further enhanced such stiffening in these mice [11].

#### 2.2.3. Potential Limitations of the Hemodynamic-Based Ultrasonography

While Doppler ultrasonography increased the understanding of blood flow within the aorta, attention needs to be taken in interpreting measurements due to the consequent reduction in the frame rate, aliasing Doppler sample volume size, and angle of incidence [58]. In addition, CD and SD may only detect changes in hemodynamics in the late stage of aortic stiffness.

Despite its wide applications in humans, there are some limitations in PWV measurement in rodents. One of the difficulties is the small time differences between the onset of Doppler flow within the two sites of the aorta in small animals [60]. Additionally, the bias may also be caused by the inaccurate estimation of the designed sites’ distances and the potential difficulty in obtaining a clear blood flow waveform of the aorta in small animals. PWV may be affected by cardiac function, which should be considered to evaluate aortic functional alterations.

### 2.3. Aortic Stiffness Assessments Based on the Tissue Motion

Tissue deformation happens before the changes of the vascular global structure and function. In recent years, the development of TDI and STI enables us to quantify tissue motion and deformation, which further altered the way that echocardiography approaches the characterization of aortic stiffness. Through the ultrasound evaluation of tissue motion, it is possible to obtain earlier and more detailed information on aortic stiffness.

#### 2.3.1. The Natures of the TDI and STI

In addition to detecting the blood flow within the aorta, Doppler signals can also be collected from tissue and recorded by TDI, either as spectral or color displays, which can be used to measure tissue stiffness with the velocities and strain rate of the tissue motion. The image acquisition of the aorta by TDI is similar between humans (Figure 8A) and rodents (Figure 8B).

STI is another newly developed ultrasonographic method used to quantify the tissue wall’s deformation of subsurface structures with high spatial resolution. The speckle pattern can be tracked consecutively from frame to frame and ultimately resolved into angle-independent 2D and three-dimensional (3D) strain-based sequences, providing both quantitative and qualitative information regarding tissue deformation and motion. So far, speckle tracking using in-vessel deformation has been limited to 2D-STI, while the application of 3D-STI in vessels is still undergoing technological developments. 2D-STI can be obtained by B-mode, and the corresponding circumferential strain rate can be calculated by using the dedicated software. Based on frame-by-frame tracking of tiny echo-dense speckles within the tissue, 2D-STI enables the calculations of motion and deformation variables, such as velocity, displacement, strain, and strain rate, through the assessment of the tissues’ lengthening and shortening [43,61]. As illustrated in the upper panel of Figure 9, with a 2D ultrasonographic image of the human aorta, a line was manually drawn along the inner side of the DAAo circumference, and the additional lines within a 15-mm-wide region of interest would be automatically generated by the software (Automated Cardiac Motion Quantificationᴬ⋅ᴵ, Qlab 13.0; Philips Healthcare, Amsterdam, The Netherlands). The shape and width of the regions of interest could be manually adjusted. A cine loop preview feature allowed visual confirmation that the internal line followed the vascular expansion and recoil movements throughout the cardiac cycle. Based on these measurements, the circumferential strain rate can be automatically or semi-automatically calculated by the software, as shown in the lower panel of Figure 9.

Due to its high feasibility and excellent reproducibility, this technique may be of particular benefit in small animal imaging. Recently, we have used 2D-STI to obtain mouse AS. As shown in Figure 10A–E, the images of the parasternal long-axis of AAo and subcostal views of DAAo were successfully obtained by B-mode imaging using a VisualSonics Vevo 3100 system (FUJIFILM VisualSonics Inc., Toronto, ON, Canada). The data were analyzed by using the Vevo Vasc software (Vevo LAB 3.2.6), including the diameter, area, strain, distensibility/elasticity, and wall thickness.

Studies have revealed several significant advantages of 2D-STI assessment over TDI. For example, the 2D-STI technique is angle-independent and is not influenced by tethering or translational motion, and the deformation pattern can be analyzed in longitudinal, radial, and circumferential directions [62]. By capturing the segmental tissue motion across multiple planes and axes serially, the analysis from the 2D-STI images could provide much greater sensitivity and specificity assessment on regional and global cardiovascular function than the conventional echocardiographic measurements [63]. Although 3D-STI has potential advantages over 2D-STI due to the third parameter of the image, particularly in evaluating cardiac pathologies [64,65], the technique and equipment requirement and the database load of 3D-STI are much more significant than 2D-STI.

#### 2.3.2. Applications of TDI and STI in Aortic Stiffness Assessments

Previous studies have revealed that TDI could be used to evaluate arterial stiffening in humans by measuring the vessel wall motion waveforms [35,66]. Multiple TDI-related parameters were applied in assessing the aortic elastic properties, including wall velocities and tissue strain (TDI-ε). The velocity could be measured at different phases, such as during systole (S′), early relaxation (E′), and atrial systole (A′) [36]. It has been shown that tissue strain was significantly impaired in diabetic patients and even more so in diabetic patients combined with hypertension [36]. The same method has been used in patients with coronary syndrome disease (CAD) [34], and the results showed that Eʹ of the anterior wall of AAo measured with TDI was decreased in patients with premature CAD and correlated with arterial stiffening [34]. TDI has also been used to measure myocardial velocities and strain rate noninvasively in mice. Both indices are sensitive markers for quantifying LV global and regional function in mice [67]. However, its application in aortic studies in animal models remains limited.

In addition, it has been demonstrated that the circumferential deformation of the aorta can be measured using 2D-STI and has been proved to be a simple and accurate determination of aortic stiffness [40,43,68,69]. Human studies have demonstrated that 2D-STI measurements are the most valuable to determine the early alteration of aortic tissue mechanical properties [26]. For example, it has been shown that the global circumferential AS could be easily assessed in patients with moderate-to-severe aortic stenosis [40]. Using 2D-STI, the AoA strain and strain rate were significantly lower in hypertensive patients than healthy individuals, indicating that 2D-STI might serve as a new and reliable approach in diagnosing vascular diseases at the early stage [43]. This method has also been used to derive biomechanical information and stratify risk in patients with hypertension, diastolic dysfunction, LV hypertrophy, atrial fibrillation, and other CAD [40,43,70].

Furthermore, 3D-STI provides a powerful tool to measure the dynamic cyclic deformation of the aorta in vivo. For example, using 3D-STI, the results found that, compared with healthy DAAo, patients with abdominal aortic aneurysms not only showed reduced mean strains but also revealed an increased spatial heterogeneity and more pronounced temporal dysynchrony delayed systole [68]. 3D-STI has also been used and validated in large animals [71]. This method is expected to provide a valuable tool to measure more detailed information related to the small and regional alterations during the early stage of vascular diseases in rodent models. Due to their advantages, these tissue motion-based techniques provide essential research tools with great potential in future clinical and research applications.

#### 2.3.3. Potential Limitations of TDI and STI

Despite the promising advantages of TDI, most of the current studies are concentrated in the myocardial structural/functional assessment, while the application in the aorta remains limited, particularly in mice, which may be due to the relative challenge in image acquisition of the echocardiogram compared to the heart [6,72].

Some limitations of 2D-STI for assessing AS are also noticed, particularly in mice. In addition to being time-consuming and the requirement of the STI software [43], the adoption of strain analysis in small animal models has been limited primarily because of technical differences in imaging mice versus humans, including limited ultrasonographic views, translational motion during image acquisition, and the effect of very high heart rates [73].

## 3. Conclusions and Future Directions

Ultrasonography is an essential non-invasive imaging tool that would offer structural and functional evaluations in the aorta and has been widely used in clinical and basic research. Although various ultrasound techniques have been developed, each has its advantages and limitations in measuring aortic stiffness. For example, structure-based assessments provide a fundamental association between pathological alteration with aortic stiffness. However, since the measurements rely on only one plane imaging or only a straight line in a direction, the parameters, such as diameter and wall thickness, can be quite variable due to the acoustic window and may not reflect the systemic functional alteration temporally. Reciprocally, although hemodynamics-based assessments reflect aortic functional alteration via measuring the velocity of blood flow, it is notable that the Doppler signals are angle-dependent, and the blood flow velocity can be affected by other factors, such as heart rate, LV ejection fraction, and regional variation along the aortic tree. Newly developed measurements on the tissue motion reflect the tissue deformation of the aortic wall, providing a more direct assessment of the aortic stiffness. It also led to an early diagnosis since the tissue deformation happens before the vascular global structural and functional changes. However, the application of these techniques may be limited by their requirement of high-quality images and time costs, and the bias caused by the variability of the analysis software. Integrating multiple ultrasonography imaging-based techniques would provide a comprehensive characterization of the vessel mechanics and improve our accurate information on the pathophysiology of aortic disease.

The applications of ultrasonography in assessing aortic stiffness are also different between humans and rodents. For example, as PWV was widely used in humans to assess aortic stiffness, its application in rodents is dramatically limited by the potential inaccuracy in measuring the time difference between the onsets of Doppler flow due to the small body size and the high heart rates of the rodents. In addition, while the ultrasonographic measurements were performed in humans in a conscious condition, they have to be taken in rodents under anesthesia; the bias may also be caused by the anesthesia in structure-basis aortic stiffness measurements due to the inconsistent alterations between the structure and BP. Furthermore, although the TDI and STI provide a valuable tool to measure the early regional alterations of aortic stiffness in humans, their applications in rodents are far behind due to the technique proving to be challenging in small animals.

Rodent models allow the ultrasonographic measurements combined with invasive methods, genetic and pharmacological inventions, and in vitro studies, providing valuable information that cannot be obtained from human studies. One primary direction in the future research of aortic diseases by using ultrasonography is to develop new techniques to overcome the current challenges in assessing aortic stiffness in these small animals. First, increasing new techniques have been invented and applied in clinical patients. These techniques could be transferred from humans to animal models for basic science research by improving the image quality and the analysis software that are applicable for small animals, such as TDI and STI. Secondly, it is crucial to take advantage of the animal models to identify the relationship between physiological alterations to the mechanistic outcomes. For example, exploring the biomarkers that can be detected by ultrasonography would improve the assessment of the hemodynamics or PWV changes in small animals. In addition, using bio-labeling techniques to trace the cell/tissue-specific targets would enhance the sensitivity and specificity of ultrasonography in measuring the tissue motion or deformation. Furthermore, using a combined continuous BP measurement and ultrasonography to detect the real-time alterations of aortic structure with the BP change in a conscious condition would provide more faithful aortic stiffness and avoid the influence of anesthesia on hemodynamics. Finally, ultrasonography can also be combined with new drugs to assess aortic stiffness in small animals. By measuring the aorta’s response to these newly developed vaso-activators or dilators, aortic stiffness could be calculated based on the drug-induced structure–BP response in vivo. These responses can be further confirmed ex vivo in animal tissues, which will, in turn, improve the in vivo measurements.

On the other hand, basic research on rodent models could bring new mechanistic information to clinicians and develop new techniques with the molecular/cell/tissue-specific targets to explore disease- or drug-related dynamic alterations. More advanced imaging technologies are bound to be used in the clinical evaluation of aortic conditions in the future research field, such as strain-based imaging, contrast ultrasonography, and ultrasound molecular imaging. Echo-derived aortic stiffness parameters could be combined with biomechanical parameters and histopathology to provide more clinical information. The clinicians, basic scientists, and pharmacologists need to work together to develop a new comprehensive system to detect vascular alterations in humans at cellular/subcellular or molecular levels.

## Figures and Tables

**Figure 1 diagnostics-11-00454-f001:**
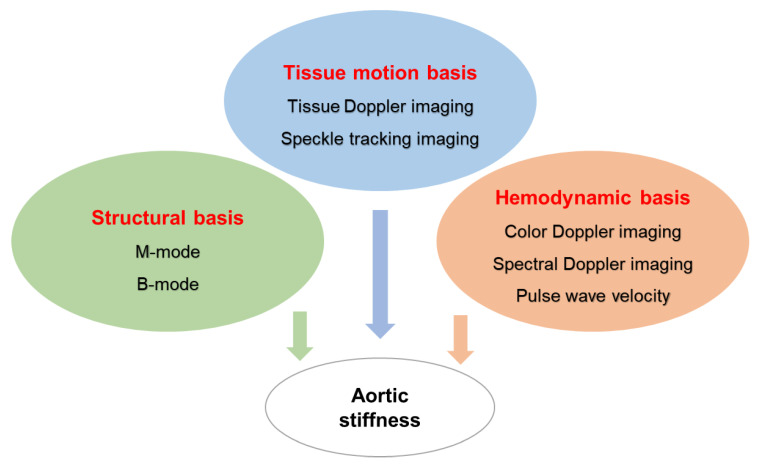
The scheme of integrating applications of ultrasonography in aortic stiffness assessments. Based on the characters of each ultrasonography, aortic stiffness could be assessed through the three major measurements: aortic structure, hemodynamics, and tissue motion.

**Figure 2 diagnostics-11-00454-f002:**
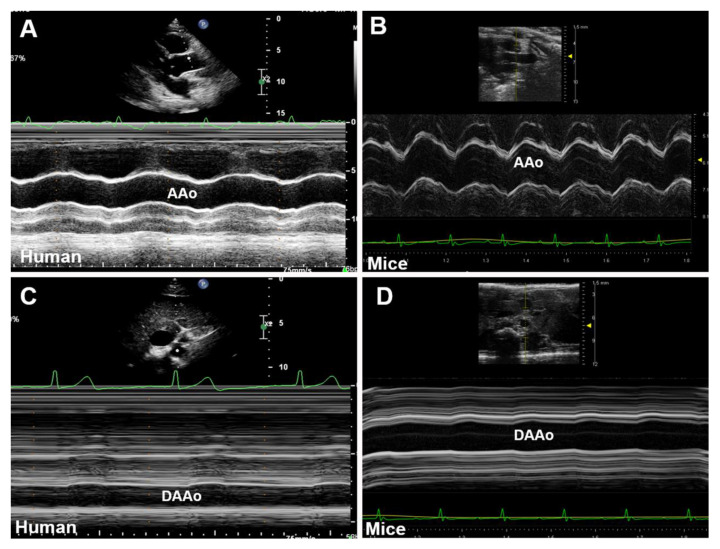
Representative M-mode displays of the aorta for the anterior–posterior diameter measurement. (**A**,**B**): The long-axis view of the ascending aorta (AAo) obtained in systole and diastole in human (**A**) and mouse (**B**). (**C**,**D**): The short-axis view of the descending abdominal aorta (DAAo) in systole and diastole in humans (**C**) and a mouse model (**D**).

**Figure 3 diagnostics-11-00454-f003:**
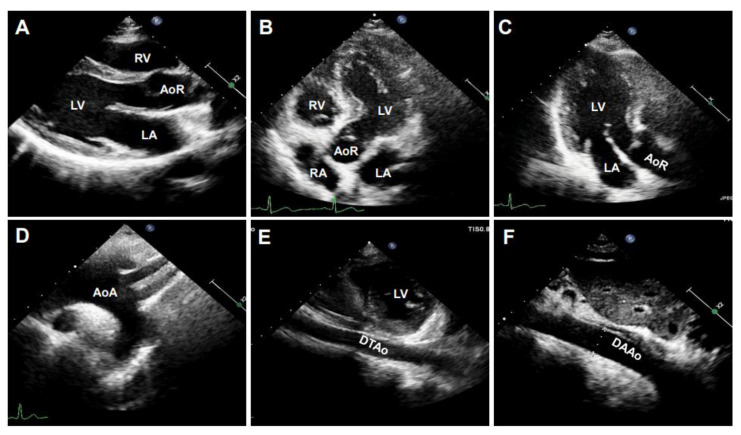
The representative images of human aorta segments were obtained by B-mode. The segments of the aorta were imaged from a 20-year-old male volunteer. (**A**): The parasternal long-axis view of the aortic root (AoR). (**B**): The apical five-chamber view of the AoR. (**C**): The apical three-chamber view of the AoR. (**D**): The suprasternal view of the aortic arch (AoA). (**E**): The parasternal view of the descending thoracic aorta (DTAo). (**F**): The subcostal view of the descending abdominal aorta (DAAo). LV: left ventricle, LA: left atrium, RV: right ventricle, RA: right atrium.

**Figure 4 diagnostics-11-00454-f004:**
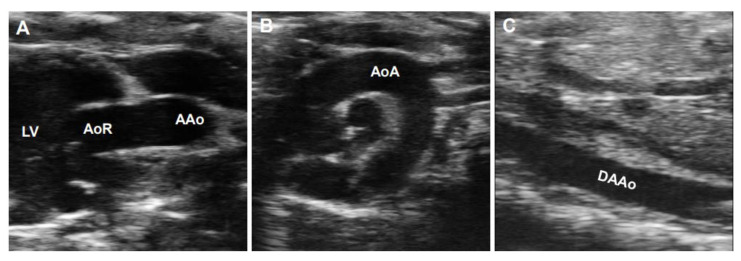
B-mode imaged segments of a mouse aorta. (**A**): The parasternal long-axis view of the aortic root (AoR) and the ascending aorta (AAo). (**B**): The suprasternal view of the aortic arch (AoA). (**C**): The subcostal view of the descending abdominal aorta (DAAo). LV: left ventricle.

**Figure 5 diagnostics-11-00454-f005:**
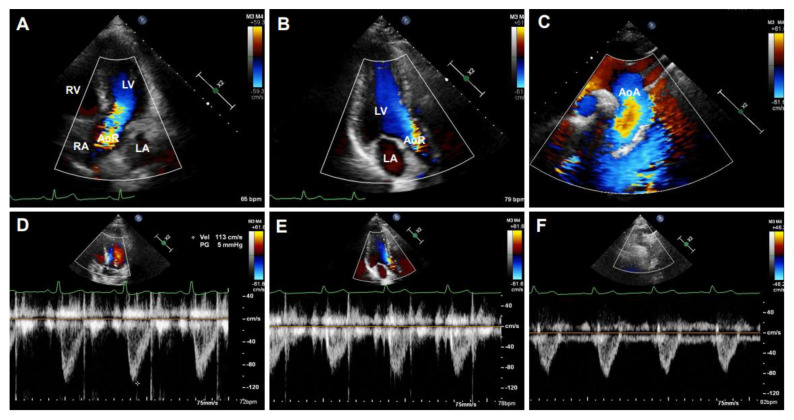
Representative Doppler signals and velocity waveforms of a human aorta. Images were obtained from a 20-year-old male volunteer representing Doppler images of the apical five-chamber view (**A**) and three-chamber view (**B**) of the aortic root (AoR) and the suprasternal view of the aortic arch (AoA) (**C**). (**D**–**F**): The corresponding assay of the aortic velocity of (**A**–**C**). LV: left ventricle, LA: left atrium, RV: right ventricle, RA: right atrium.

**Figure 6 diagnostics-11-00454-f006:**
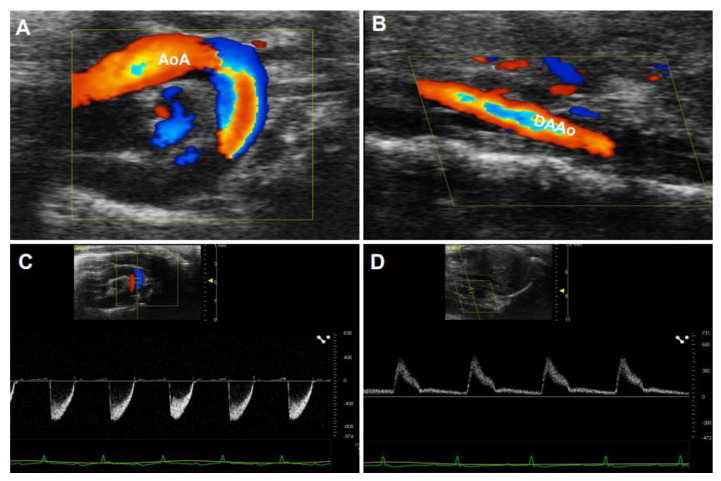
Color Doppler(CD) and pulsed-wave images from a mouse aorta. The representative images by CD were obtained from a healthy four-month-old C57BL/6J mouse. (**A**,**B**): CD images of the aorta arch (AoA) (**A**) and descending abdominal aorta (DAAo) (**B**). (**C**,**D**): The corresponding Inflow velocity of (**A**,**B**) using pulsed-wave Doppler imaging.

**Figure 7 diagnostics-11-00454-f007:**
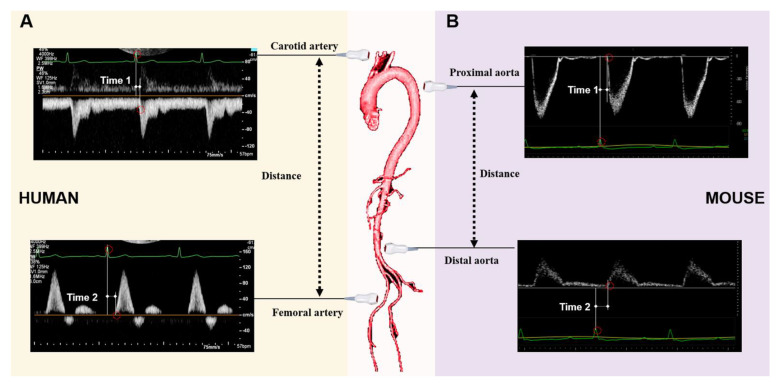
An illustrative example of pulse wave velocity measurement by ultrasonography. (**A**): The illustration of the distance and transit times assessed from signals detected at the carotid and femoral arteries in humans. (**B**): The illustration of the distance and transit times registered in the distal aortic arch and in the distal abdominal aorta in a mouse model.

**Figure 8 diagnostics-11-00454-f008:**
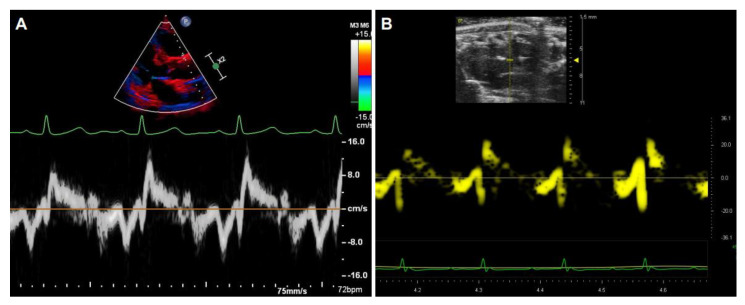
Representative images of the aorta and the waveforms by tissue Doppler imaging. The segmental velocity information of the anterior wall of the aorta in human (**A**) and mice (**B**).

**Figure 9 diagnostics-11-00454-f009:**
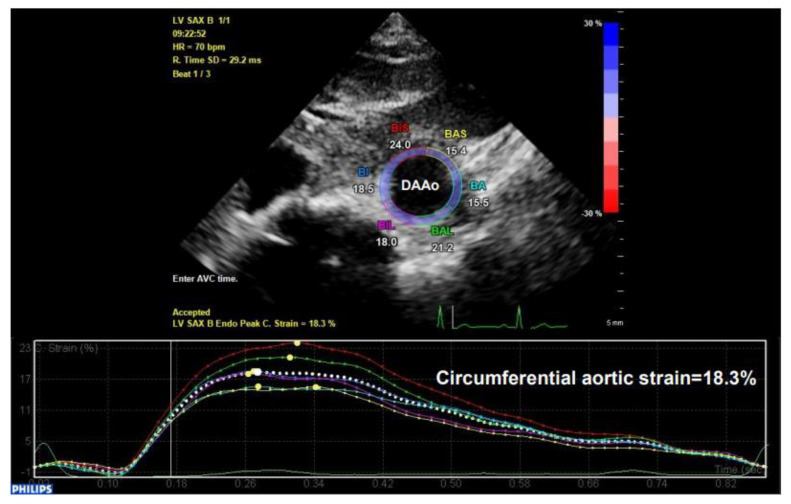
The illustration of human aortic assessments by two-dimensional (2D) speckle tracking. Intraoperative 2D speckle tracking analysis from a short-axis view of descending abdominal aorta (DAAo) in a healthy individual. The circumferential strain profile is displayed on a positively directed curve with a peak value of 18.3%.

**Figure 10 diagnostics-11-00454-f010:**
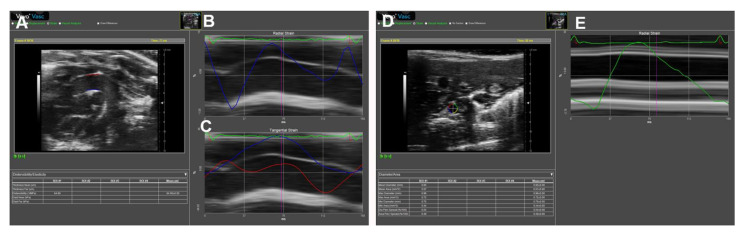
Representative images of mouse aortic assessments by 2D speckle tracking. Images acquired from a four-month-old C57BL/6J mouse over one cardiac cycle. (**A**): B-mode image of the ascending aorta (AAo) in long-axis orientation. (**B**): Regional radial strain curve superimposed M-mode image of the aorta. (**C**): Tangential strain curve superimposed M-mode image of the aorta. (**D**): A B-mode image of the descending abdominal aorta (DAAo) in short-axis orientation. (**E**): Regional radial strain curve superimposed M-mode image of the aorta.

**Table 1 diagnostics-11-00454-t001:** Formulas for calculating the biophysical properties of the aorta.

Echo Technique	Formula	References
M-mode	Stiffness index (SI) = ln (SBP − DBP)/[(AoS − AoD)/AoD]	[22,23,33,34,35,36,37,38]
	Distensibility (DI) = 2 × (AoS − AoD)/[AoD × (SBP − DBP)]	[33,34,35,36,37,38,39,40]
	M-mode strain = (AoS − AoD)/AoD	[15,22,36,37,38,39]
	Compliance= (AoS − AoD)/(SBP − DBP)	[22,36]
	Elastic pressure-strain modulus (Ep) = (SBP − DBP)/[(AoS − AoD)/AoD] Ep = 1333(SBP − DBP)/[(AoS − AoD)/AoD]	[34,41] [15,23]
	Young’s modulus (E_Y_ = AoD/thickness of aortic wall/DC)	[23]
	Peterson’s elastic modulus = AoD(SBP − DBP)/(AoS − AoD)	[36]
M-mode B-mode	Aortic strain = 100(AoS − AoD)/AoD	[19,33,37]
	β index = ln (SBP/DBP)/[(AoS − AoD)/AoD]	[5,15,39,40,41,42,43]
	Circumferential strain = 1/2 × [(AoS/AoD)^2^ − 1] × 100%	[25,31,44]

“AoS” to the aortic diameter in systole; “AoD” to the aortic diameter in diastole; “ln” refers to the natural logarithm; “SBP” to the systolic blood pressure; “DBP” to the diastolic blood pressure; “BSA” to body surface area; DC: = ΔA/A × (SBP − DBP), while A: the minimal cross-sectional area of the aorta, ΔA: the maximal minus minimal cross-sectional area of the aorta.

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
