# Peer review of "The Progress of Advanced Ultrasonography in Assessing Aortic Stiffness and the Application Discrepancy between Humans and Rodents"

_diagnostics, 2021, doi:10.3390/diagnostics11030454_

Round 1

Reviewer 1 Report

The manuscript #diagnostics-1117814 entitled “The progress of the advanced ultrasonography in assessing aortic stiffness and the application discrepancy between humans and rodents”, is of considerable interest. However, the issues reported below should be addressed to improve the quality of this review.

Major criticisms:

Conclusion and future directions section is too short. Additional prospective ideas are needed to make this a true Expert Opinion section. Please expand this section. 

Minor comment:
1) Table 2 (presently split over two pages) needs to be re-adjusted to one page.

Reviewer 2 Report

Dear Authors

Please consider these points:

1-Please revise Fig 1, as the arrows meaning and direction is not understandable. And explain more if possible.

2-some information in the text is about human and some about rodents. Please change the title or the text to be in line with each other. For example, application discrepancy between humans and rodents should be focused in the conclusion if the current title is held.

3-The last table of the conclusion is not necessary and please move it to the discussion with more detail information.

Reviewer 3 Report

It is an excellent review of the various echocardiographic methods used to evaluate aortic elasticity in humans and their potential use in animal models. The article is well written and very clear. In section "2.1. Structure-based assessment of aortic stiffness" since the authors describe the use of ultrasound in the study of structural and functional alterations of the aorta in humans and animals, it would be appropriate to cite a recent article entitled "Ultrasonographic assessment of abdominal aortic elasticity in hypertensive dogs" (J. Vet. Internal Medicine, Corda A. et al 2020) in which the authors evaluated abdominal aortic elasticity using B -mode derived Ao Strain in dogs with and without systemic hypertension.

Line 340 "Sd" should be replaced by "SD".

Reviewer 4 Report

General comment: The authors presented an interesting review work concerning to the progress of ultrasonography in assessing aortic stiffness and its different application between humans and rodents.

The manuscript is well structured and written in a comprehensive way.

Title: The title is adequate.

Abstract: It is adequate. The keywords should be different from those used in the title.

Introduction: The Introduction is adequate.

Line 92: “…including a). Real-time B-mode images and M-mode traces”. Please delete the dot after a)

Lines 121-122: The reference to the images should be replaces as follow “…and its application in assessing aortic dimensions are similar between humans (Fig.2A and C) 121 and rodents (Fig.2B and D).

Table 1: The authors should reorganize the table, presenting first the formular for M-mode and then the formular for M-mode and B-mode. In this way, the authors avoid text repetition in the first column and the table will become more “clean”.

The abbreviations should be defined in its first use and used throughout the manuscript. Please check.

References: They are adequate.

Recommendation: The manuscript should be accepted for publication after a Minor revision.

Round 2

Reviewer 1 Report

none.

Reviewer 2 Report

Thank you for your revision

the questions been answered